# A Novel Method for the Early Detection of Single Circulating, Metastatic and Self-Seeding Cancer Cells in Orthotopic Breast Cancer Mouse Models

**DOI:** 10.3390/cells13141166

**Published:** 2024-07-09

**Authors:** Muhammad Murad, Yanjiang Chen, Josephine Iaria, Adilson Fonseca Teixeira, Hong-Jian Zhu

**Affiliations:** 1Department of Surgery, The Royal Melbourne Hospital, The University of Melbourne, 5th Floor Clinical Sciences Building, Parkville, VIC 3050, Australia; muhammadmura@student.unimelb.edu.au (M.M.); yanjiang.chen@usz.ch (Y.C.); jiaria@unimelb.edu.au (J.I.); afonsecateix@student.unimelb.edu.au (A.F.T.); 2Huagene Institute, Kecheng Science and Technology Park, Pukou District, Nanjing 211806, China

**Keywords:** cancer, luciferase assay, bioluminescence imaging, circulating tumour cells, metastasis, tumour self-seeding, mouse model

## Abstract

Background: Metastasis is the main cause of cancer-related deaths, but efficient targeted therapies against metastasis are still missing. Major gaps exist in our understanding of the metastatic cascade, as existing methods cannot combine sensitivity, robustness, and practicality to dissect cancer progression. Addressing this issue requires improved strategies to distinguish early metastatic colonization from metastatic outgrowth. Methods: Luciferase-labelled MDA-MB-231, MCF7, and 4T1 breast cancer cells were spiked into samples from tumour-naïve mice to establish the limit of detection for disseminated tumour cells. Luciferase-labelled breast cancer cells (±unlabelled cancer-associated fibroblasts; CAFs) were orthotopically implanted in immunocompromised mice. An ex vivo luciferase assay was used to quantify tumour cell dissemination. Results: In vitro luciferase assay confirmed a linear and positive correlation between cancer cell numbers and the bioluminescence detected at single cell level in blood, brain, lung, liver, and mammary fat pad samples. Remarkably, single luciferase-labelled cancer cells were detectable in all of these sites, as the bioluminescence quantified in the analysed samples was substantially higher than background levels. Ex vivo, circulating tumour cells, metastasis, and tumour self-seeding were detected in all samples from animals implanted with highly metastatic luciferase-labelled MDA-MB-231 cells. In turn, detection of poorly metastatic luciferase-labelled MCF7 cells was scarce but significantly enhanced upon co-implantation with CAFs as early as 20 days after the experiment was initiated. Conclusions: These results demonstrate the feasibility of using an ultrasensitive luciferase-based method to dissect the mechanisms of early metastatic colonization to improving the development of antimetastatic therapies.

## 1. Introduction

Metastasis is the main cause of cancer-related deaths [1,2]. During cancer progression, cancer cells undergo several steps to successfully disseminate and establish metastatic lesions. This includes invasion to surrounding tissues, intravasation into the vasculature, survival in the circulation, extravasation in distant sites, re-seeding, and the growth of secondary tumours [3]. Despite the clinical relevance of metastasis, current metastasis models exhibit critical limitations that compromise the precise and comprehensive characterization of the mechanisms underlying this outcome [4]. Consequently, while the development of therapies targeting primary tumours has shown clear progress over the years, the field of antimetastatic therapies still lags behind [5,6].

Different from in vitro cancer models, in vivo models of cancer metastasis are characterized by systemic effects that better recapitulate real human cancers [7]. In addition to autochthonous mouse models, ectopic and orthotopic implantation of cancer cells have been widely used to study metastasis [7,8,9]. As the primary tumour progresses, cancer cells may invade surrounding tissues, which has long been assessed by histological analyses of resected tumours [7,10]. Yet, as invasive cancer cells undergo a series of challenges before reaching metastatic niches, other molecular and phenotypic changes are required for the establishment of metastases [6,11,12]. A comprehensive understanding of the metastatic cascade requires investigating the steps that follow invasion, especially cancer cell intravasation, extravasation, re-seeding, and metastasis outgrowth. 

The investigation of cancer cells moving to and from the lumen of mouse blood vessels may be achieved by intravital microscopy [13,14,15,16]. Nevertheless, intravital microscopy analyses are limited by factors such as the selection of only one or a few regions of interest (not necessarily comparable across different animals) and reduced light penetration depth, which decrease the potential of this method for quantitative analyses [17,18]. Alternatively, cancer cell intravasation and extravasation may be investigated by quantifying circulating tumour cells (CTCs). Many methods used for CTC quantification rely on the isolation of these cells by physical properties or immunocapture [19,20]. These strategies compromise the use of such methods at earlier stages of cancer progression when only a few CTCs exist at a given time. Due to its superior sensitivity, quantitative polymerase chain reaction (qPCR) has been often used to overcome this issue [21,22], but extensive sample processing makes this method time-consuming [23]. Moreover, the analysis of rare samples is commonly associated with stochastic cDNA amplification and increases errors introduced during sample preparation [23]. Altogether, these variables reduce both the accuracy of this method and the practical use of qPCR for the quantification of CTCs in large cohorts.

Similar problems are observed for the detection and quantification of metastatic cancer cells by intravital microscopy or qPCR [22,24,25,26,27]. In vivo bioluminescence imaging (BLI) helps to accomplish this goal as it enables monitoring the spread of luciferase-labelled cancer cells in mouse cancer models [28,29,30,31]. In this method, luciferase enzymes expressed by labelled cells emit light after substrate oxidation, which is detected by an in vivo imaging system (IVIS). Although researchers have dedicated great efforts to engineer luciferase mutants, reduce the signal-to-background ratio, and improve IVIS, in vivo BLI still needs to be combined with other methods as the detection of single cancer cells cannot yet be achieved in most metastatic organs [22,32,33,34]. This also forces many researchers to rely on the analysis of advanced macrometastatic lesions [7]. However, metastasis outgrowth may better indicate the adaptation and proliferative potential of cancer cells in a secondary site rather than the ability of these cells to progress along the entire metastatic cascade [4,35]. Altogether, such shortcomings negatively affect the precise understanding of the molecular forces driving metastasis, as current methods cannot be used to investigate early metastatic events—especially those driving the re-seeding of CTCs and the initial colonization of secondary sites [4,6,35]. 

Our group has recently reported a strategy to overcome this methodological barrier by combining the luciferase labelling of cancer cells with ex vivo luciferase analysis. This strategy was proved feasible and further expanded to establish an orthotopic glioblastoma model that allows the systemic detection of single luciferase-labelled cancer cells [36]. Here, we build on these findings to establish a novel, robust, and fast method that enables the analysis of breast cancer metastases early after the re-seeding of disseminated cancer cells into metastatic niches. The results presented here pave the way for a clear understanding of the molecular mechanisms driving cancer progression and the evaluation of novel therapeutic strategies capable of impairing metastasis in different types of cancer.

## 2. Materials and Methods

### 2.1. Cell Lines and Cell Culture

The human MDA-MB-231 and MCF7 breast cancer cell lines and the murine 4T1 breast cancer cell line were purchased from the American Type Culture Collection (ATCC, Manassas, VA, USA). The human 19TT breast cancer-associated fibroblast cell line was kindly provided by Professor Peter ten Dijke. Cell lines were cultured in Dulbecco’s modified Eagle’s medium DMEM (Thermo Fisher Scientific, Waltham, MA, USA) supplemented with 5% foetal calf serum (HyCloneTM, GE Healthcare Life Sciences, Chicago, IL, USA), 10 µg/mL penicillin, and 100 µg/mL streptomycin (Invitrogen, Thermo Fisher Scientific, Waltham, MA, USA). Cell lines were maintained at 37 °C with 10% CO_2_ supply in a humidified atmosphere.

### 2.2. Establishing Stable Luciferase-Labelled Cell Lines

Human MDA-MB-231 and MCF7 cell lines were stably transfected with pCMV-KDEL-Gluc1 DNA plasmid vector. Murine 4T1 cells were transfected with pGL4.51[luc2/CMV/Neo] DNA plasmid vector. Plasmid transfection used Fugene^®^ HD (Promega, Madison, WI, USA). Transfected cells were selected by 2 mg/mL G418 sulfate (Roche, Basel, Switzerland). This concentration of G418 sulfate was established by treating breast cancer cell lines with increasing concentrations of G418 sulfate and visually determining the minimum concentration of this antibiotic that could kill all the cells after 72 h in an antibiotic kill curve. Constitutive luciferase expression was confirmed by luciferase assay in cells resistant to G418 sulfate after clone selection. Luciferase-labelled cells were designated MDA.Gluc (Gaussia luciferase-labelled MDA-MB-231 cells), MCF7.Gluc (Gaussia luciferase-labelled MCF7 cells), and 4T1.Fluc (Firefly luciferase-labelled 4T1 cells).

### 2.3. Luciferase Activity Quantification and Cell Titration

Adherent cells were trypsinized, centrifuged (300× *g*), and resuspended in PBS (1X). Cell numbers were established by direct counting using a Neubauer chamber. Cell suspensions were then diluted to obtain eight dilution points ranging from 1 to 128 cells per 50 μL PBS (1X). Cells were lysed using Cell Culture Lyses Reagent (Promega, Madison, WI, USA) (30 min; 4 °C) and the luciferase activity was quantified in a 96-well opaque reading plate. A GloMax^®^ 96 Microplate Luminometer (Promega, Madison, WI, USA) was used to detect and quantify the bioluminescence resulting from automated injection of Luciferase Assay Kit (Promega, Madison, WI, USA). Luciferase activity is represented throughout this study as relative luciferase units (RLU).

### 2.4. In Vivo Experiments

All animal experiments were performed according to the National Health Medical Research Council of Australia code as approved by The University of Melbourne Animal Ethics Committee (Ethics ID: 1613813.1). Six-to-eight weeks old female severe combined immune-deficient (NOD-SCID) mice were purchased from the Animal Resource Centre (ARC, Perth, Australia). Only female mice were used in this study as to evaluate the biology of human female breast cancers. Mice were orthotopically inoculated in contralateral inguinal fat pads (4th pair of mammary glands) with 3 × 10^6^ luciferase-labelled MDA-MB-231 cells and 1 × 10^6^ wild type (unlabelled) MDA-MB-231 cells, resuspended in 100 μL of PBS. Using the same protocol, mice were alternatively implanted with 5 × 10^6^ luciferase-labelled MCF7 cells (±1.5 × 10^6^ CAFs) and 5 × 10^6^ wild type (unlabelled) MCF7 cells. Different numbers of luciferase-labelled and unlabelled cancer cells were used to balance proliferation and cancer growth. For this procedure, mice were anaesthetised by isoflurane gas inhalation via nose cone attached to Xenogen XGI-8 Anesthesia System (Caliper Life Sciences, Hopkinton, MA, USA). Cancer cells were inoculated by using syringes connected with 27 G needles. The experimental endpoint was defined as 28 days post-inoculation for analysis involving MDA-MB-231 cells or 20 days post-inoculation for analysis involving MCF7 cells. Following experimental endpoint, mice were humanely killed by CO_2_ inhalation. Approximately 600–800 μL of blood was drawn via cardiac puncture through the diaphragm using syringes connected with 23 G needles and collected in microtubes containing heparin to prevent clotting. Tumours and organs required for further analysis were excised, washed with PBS (1X), and stored at −20 °C. 

### 2.5. Ex Vivo Luciferase Assay for Detection and Quantification of Luciferase-Labelled Tumor Cells

Luciferase activity was quantified in lysates from blood, organs, and tumours harvested from mice inoculated with luciferase-labelled tumour cells. In brief, mouse blood samples were mixed with Cell Culture Lyses Reagent (Promega, Madison, WI, USA) for cell lysis (30 min; 4 °C) in a proportion of 5 µL blood to each 25 µL lysis buffer. Next, 30 μL of lysate was transferred per well into a 96-well opaque reading plate, and the luciferase activity was quantified in a GloMax^®^ 96 Microplate Luminometer (Promega, Madison, WI, USA) following the automated injection of Luciferase Assay Kit (Promega, Madison, WI, USA). Additionally, frozen organs and tumours were mechanically dissociated, and obtained fragments were randomly selected and grounded by using a manual pestle and mortar on dry ice. Processed fragments were weighed and 25 μL Cell Culture Lyses Reagent (Promega, Madison, WI, USA) were added per 5 mg of processed sample. Samples were lysed (30 min; 4 °C) and lysates were transferred into a 96-well opaque reading plate for luciferase activity quantification. Luciferase activity is represented throughout this study as relative luciferase units (RLU). Normalization of the bioluminescence into cell numbers was carried out by interpolating raw RLU quantified by ex vivo luciferase assay in the standard curve established in this study and presented in the results section. RLU quantified in blank samples (i.e., solution not containing luciferase-labelled cells) was subtracted from RLU quantified in mouse samples to account for background noise. 

### 2.6. Statistical Analysis

Statistical analysis was performed using GraphPad Prism version number 8.0.2 (GraphPad Software Inc., San Diego, CA, USA). Correlations between luciferase activity and cell numbers were analysed by best-fit linear regression analysis. Groups were compared by a two-sided unpaired Student’s *t*-test. In vitro and ex vivo results represent mean ± standard error of the mean (SEM). Differences between groups were considered statistically significant when *p* < 0.05.

## 3. Results

### 3.1. Luciferase Activity Is Detected In Vitro in Single Luciferase-Labelled Tumour Cells

Luciferase enzymes have been widely used to label immortalized cell lines [28,36,37,38,39,40,41]. Here, we aimed at developing a highly sensitive and robust method for the detection and quantification of single tumour cells. To this end, we stably transfected human breast cancer cells (MDA-MB-231 and MCF7 cell lines) with pCMV-Gluc-KDEL plasmid vector and murine mammary carcinoma cells (4T1 cell line) with pGL4.51[luc2/CMV/Neo] plasmid vector. Successful transfection was confirmed by clonal selection. Specifically, single cells were seeded in 96-well plates (0.5 cell/well), clones were grown independently and evaluated for luciferase activity. Clones with the highest luciferase activity were selected, expanded, and used in the following analyses. Next, we quantified the luciferase activity per cell to evaluate the repeatability and sensitivity of our method. Luciferase-labelled cells resuspended in PBS were serially diluted to a single cell and lysed, and the luciferase activity was quantified in vitro. As expected, a strong positive correlation was observed between cell numbers and luciferase activity for MDA.Gluc (Figure 1A), MCF7.Gluc (Figure 1B), and 4T1.Fluc (Figure 1C) cells. Moreover, as observed for all cell lines evaluated, the luminescence quantified in wells containing a single luciferase-labelled cell was significantly higher than that detected in blank solution (background) (Figure 1). These results confirm the reproducibility, sensitivity, and accuracy of our method for the detection and quantification of luciferase-labelled tumour cells in vitro. 

### 3.2. Tumour Cells Spiked in Mouse Blood and Organ Samples Are Detected at Single Cell Level 

Next, we sought to confirm the efficiency of our method by evaluating its ability to detect and quantify luciferase-labelled tumour cells among non-labelled non-tumour cells. Specifically, MDA.Gluc, MCF7.Gluc, and 4T1.Fluc cells were spiked in blood, brain, lung, liver, and mammary fat pad samples harvested from tumour-naïve NOD-SCID mice to represent mocking models for CTCs, multiorgan metastasis, and tumour self-seeding. First, mouse samples were processed, and their volume (blood) and weight (brain, lung, liver, and mammary fat pads) quantified. Following this procedure, solutions containing luciferase-labelled cancer cells were serially diluted, and a set number of labelled cells were spiked in mice samples. After lysis, the luciferase activity was quantified in sample lysates, and RLU were normalized per volume (5 μL of blood sample) or weight (5 mg of brain, lung, liver, and mammary fat pads). As observed for the titration of luciferase-labelled cells resuspended in PBS (Figure 1), the luminescence detected in blood samples was directly and strongly correlated with increased number of spiked MDA.Gluc (Figure 2A), MCF7.Gluc (Figure 2B), and 4T1.Fluc (Figure 2C) cells. Additionally, single luciferase-labelled tumour cells spiked in 5 μL of blood samples also emitted significantly higher luminescence than that attributed to background levels (Figure 2), reinforcing the sensitivity of the method and its potential applicability for the quantification of CTCs. 

Similar results were observed when MDA.Gluc, MCF7.Gluc, and 4T1.Fluc cells were spiked in organ samples. The bioluminescence quantified in brain (Appendix A), lung (Appendix A), liver (Appendix A), and mammary fat pad (Appendix A) samples harvested from tumour-naïve mice was directly and strongly correlated with the number of spiked luciferase-labelled cells. These results establish the feasibility of this method for the detection and quantification of single metastatic cells in multiple sites critically impacted by breast cancer metastasis [42,43]. Additionally, it substantially improves the strategies currently used to investigate tumour self-seeding, another outcome associated with cancer progression [37,38,44,45].

### 3.3. Luciferase-Labelling Enables the Detection of CTCs, Multiorgan Metastasis and Tumour Self-Seeding in an MDA-MB-231 Orthotopic Breast Cancer Model

After confirming the elevated sensitivity of our method and developing a standard curve that represents the correlation between the luciferase activity and the number of cells spiked into blood, brain, lung, liver, and mammary fat pad samples, we next sought to validate this method in vivo. To achieve this aim, highly metastatic MDA-MB-231 cells were used as a model. Specifically, 3 × 10^6^ MDA.Gluc cells and 1 × 10^6^ wild type (unlabelled) MBA-MB-231 cells were contralaterally implanted into the mammary fat pads of NOD-SCID mice (Figure 3A). Twenty-eight days post-tumour inoculation, animals were humanely killed, blood and organ samples were harvested, and the luciferase activity was quantified in sample lysates. Cell numbers and experiment duration were defined considering our previous studies [37,38] to compensate for the differences in the growth of wild type and luciferase-labelled cells. Remarkably, luciferase activity was detected above background levels in all samples analysed (Figure 3B), conclusively demonstrating the progression of primary tumours towards invasion, intravasation, extravasation, and colonization of secondary organs. Furthermore, normalization of the RLU quantified in each blood sample analysed showed a range of 3–40 CTCs per 5 μL of blood (Figure 3B), confirming the rarity levels of such a cell population as previously described [21,22,46]. Interestingly, a similar range of luciferase-labelled cells was also detected in brain samples (6–34 cells per 5mg of brain sample) (Figure 3B). These results highlight that metastatic MDA-MB-231 breast cancer cells are capable of re-seeding and colonizing this organ quickly after primary tumour formation, as also reported by our group in previous studies [37,38]. Lung samples, however, were abundantly colonized by metastatic cells, and a range of 20–235 luciferase-labelled MDA-MB-231 cells were detected per 5 mg of lung (Figure 3B). Surprisingly, although the liver is typically ranked among the main sites targeted by breast cancer metastasis [47,48,49], only 2–14 tumour cells were detected per 5 mg of liver sample (Figure 3B). Conversely, tumour self-seeding was confirmed as a major event in this orthotopic breast cancer model, as 467–9290 luciferase-labelled tumour cells were quantified per 5 mg of unlabelled tumour (Figure 3B). These results establish the feasibility of using our method for the investigation of cancer progression as we provide a robust model for the analysis of the early stages of the metastatic cascade, thus addressing a critical issue in existing mouse cancer models.

### 3.4. Luciferase-Labelling Allows Ex Vivo Quantification of the Dissemination of Poorly Metastatic MCF7 Breast Cancer Cells

Although some cancer cell lines are capable of quickly forming highly aggressive tumours if orthotopically implanted (e.g., MDA-MB-231 cells), other cancer cell lines may require support from the tumour microenvironment (TME). CAFs are abundant components of the TME that have been critically associated with cancer progression [15,50]. Due to CAFs relevance to cancer metastasis, an increasing number of studies have focused on investigating their role in this process [50,51]. To further assess the value of our method and validate its use in a different breast cancer model, 5 × 10^6^ MCF7.Gluc cells were orthotopically co-implanted with or without 1.5 × 10^6^ wild type (unlabelled) CAFs into the mammary fat pads of NOD-SCID mice (Figure 4A). Additionally, 5 × 10^6^ wild type (unlabelled) MCF7 cells were contralaterally implanted for the quantification of tumour self-seeding levels (Figure 4A). Mice were monitored and humanely culled 20 days post-tumour implantation for the harvesting of blood, organ, and unlabelled tumour samples (Figure 4A). The duration of this experiment and the number of cells to be orthotopically inoculated considered our previous experience [37,38] and were defined to compensate for the growth of labelled and unlabelled tumours. Harvested samples were processed for ex vivo quantification of the luciferase activity, and raw RLU were normalised into cell numbers equivalent to CTCs, metastasis, and tumour self-seeding. 

Validating the relevance of our method, we detected Gaussia luciferase activity significantly higher than background levels in all sites analysed (Figure 4B–F). Interestingly, whereas the number of CTCs (Figure 4B) and lung metastases (Figure 4C) were not impacted by CAFs, our results demonstrate that MCF7.Gluc cells are capable of intravasating and metastasizing soon after tumour implantation. To some extent, this observation contradicts the general assumption that MCF7 cells are poorly metastatic cancer cells [52]. Furthermore, co-implantation with CAFs significantly increased the spreading of MCF7.Gluc cells to the brain (Figure 4D) and liver (Figure 4E). Similarly, whereas MCF7.Gluc cells were detected in over 30% of the unlabelled tumours harvested from control mice not implanted with CAFs, co-implantation with CAFs substantially enhanced tumour self-seeding (Figure 4F). These results reinforce the importance of using an ultrasensitive method capable of detecting the dissemination of single cancer cells quickly after tumour implantation. 

## 4. Discussion

Metastasis is a multistep process, and it is responsible for most cancer-related fatalities [53,54]. Although several therapeutic strategies have been evaluated in pre-clinical cancer models and reported to successfully impair this outcome, existing therapeutics cannot prevent metastasis or treat secondary tumours in human patients [5,6,12]. Bridging this gap requires improved models of spontaneous metastasis and highly sensitive methods capable of detecting single metastatic cells [5]. This combination is needed to differentiate the molecular mechanisms that underlie the colonization of secondary sites from those driving the outgrowth of metastatic lesions [4,7]. Here, we benefit from the elevated activity exhibited by luciferase enzymes to establish an ultrasensitive method that enables the detection of luciferase-labelled cancer cells at the single cell level. Additionally, we demonstrate the feasibility of quantifying luciferase-labelled cancer cells in secondary sites by showing a robust and linear correlation between increasing cancer cell numbers and luciferase activity analysed in multiple tissue and organ lysates. Validating the reproducibility of this method, we report similar results across different cancer cell lines and confirm high efficiency using either Gaussia luciferase labelling or Firefly luciferase labelling. Moreover, we establish novel models for the study of spontaneous breast cancer metastasis. Our model is based on the orthotopic implantation of luciferase-labelled breast cancer cells and allows fast quantification of CTCs, metastasis, and tumour self-seeding early after metastatic dissemination. Noteworthy, such results were achieved only 20 days after the simple inoculation of cancer cells resuspended in saline solution, exhibiting great potential to be further optimized by using a Matrigel solution that would offer additional support to cancer cells at the time of the implantation. 

The successful establishment of metastases involves a sequence of events that includes the invasion of cancer cells at the primary site, their dissemination, the colonization of secondary sites, and the outgrowth of metastatic lesions. Although this study focused on investigating blood-borne metastases, it is important to note that the intravasation and dissemination of cancer cells via lymph vessels may also account for a significant part of the metastatic lesions, and other organs (e.g., lymph nodes) must be investigated in future studies. Proper modelling of cancer cell re-seeding and colonization of metastatic niches in pre-clinical cancer models have been mostly restricted by poor sensitivity and time-consuming methods [55,56]. Conversely, by simple and quick sample processing, our method detects single metastatic cells less than a month after the orthotopic implantation of luciferase-labelled breast cancer cells into the mammary fat pads of female NOD-SCID mice. The relevance of this method is proven by demonstrating that while some sites are colonized by hundreds-to-thousands of breast cancer cells, other sites exhibit only a few cancer cells within the same period, which complicates the detection of early metastasis. Moreover, as organs targeted by metastatic colonization vary according to the cancer model used, acknowledging this difference is crucial to determining the efficacy of antimetastatic therapies in pre-clinical models. As seen in this study, MDA-MB-231 cells exhibit elevated levels of spontaneous metastasis to lungs and low levels to liver. In turn, MCF7 cells show a nearly inverse pattern of metastatic tropism, especially if co-implanted with CAFs. Yet, because MDA-MB-231 cells are a common model of breast cancer progression, the lungs are the only metastatic niche analysed in many studies, while changes in the metastatic colonization of other organs are often ignored. Thus, establishing and implementing a consistent, sensitive, and quick method to detect single metastatic cells in multiple organs is paramount to investigating the molecular forces that drive cancer progression and the efficacy of antimetastatic therapies against different cancer cell lines.

Our analyses also show that a high heterogenous distribution of metastatic cells is a common feature across all sites analysed. In other words, some regions of an organ that is colonized by metastatic cells may contain 5–10-fold more metastatic cells than other regions of the same organ, which are free from metastatic cells in some cases. This may not represent a problem for histological analyses or in vivo BLI if these methods are exclusively used to examine metastatic sites once they were heavily colonized by metastatic cells or after the establishment of macrometastatic lesions [7,10]. Nevertheless, it certainly poses a problem for studies focused on investigating earlier days of metastatic colonization when only a few cancer cells re-seeded into these organs. Therefore, our results show that the use of previously described methods may lead to false negative results. In this context, some sites, such as the liver, would be neglected as a niche for cancer cell lines such as MDA-MB-231 cells, while lungs would be neglected for models with tropism similar to that of MCF7 cells, and the brain would be neglected in both circumstances. 

In addition to proving the feasibility of using luciferase-labelled MDA-MB-231 and MCF7 cells as models of spontaneous breast cancer metastasis, our study also indicates the feasibility of similarly using other luciferase-labelled cell lines. As seen for MDA-MB-231 and MCF7 cells, luciferase-labelled 4T1 breast cancer cells spiked into blood, brain, lung, liver, and mammary fat pad samples were also detectable at the single-cell level. Noteworthy, the metastatic dissemination of 4T1 cells was not investigated in vivo in the current study. However, this effort in expanding the diversity of existing models of cancer metastasis may enable future studies that aim at investigating breast cancers that are distinguished, for example, by metastatic potential and response to immune therapies. In fact, our group has previously used MDA-MB-231 and MCF7 breast cancer cell models to investigate the role played by pro-metastatic signaling pathways and develop novel therapeutic strategies to prevent metastatic dissemination [37,38]. Our previous studies demonstrated that luciferase-labelled MDA-MB-231 cells exhibit elevated aggressiveness, colonizing multiple organs as quickly as 25 days after orthotopic implantation due to exosome-mediated TGF-β signaling hyperactivation and inhibition of BMP signaling [37,38]. MCF7 cells, nevertheless, are generally considered poorly invasive breast cancer cells [52]. Yet, by orthotopically implanting luciferase-labelled MCF7 breast cancer cells in a model of spontaneous metastasis, we were able to detect lung metastasis even when primary tumours were still smaller than 50 mm^3^ [38]. Additionally, we showed that exogenously supplying exosomal TGF-β to MCF7 breast cancer cells enhances lung metastasis, promotes the metastatic colonization of the liver, bones, and brain, and enables tumour self-seeding [38]. Moreover, as reported here for breast cancer cell lines, we have also shown that luciferase-labelled U87MG and MU20 glioblastoma cells are similarly detectable at the single-cell level [36]. Consistent with this observation, we established that both U87MG and MU20 cells may be used as novel models to investigate glioblastoma recurrence and systemic dissemination if labelled with luciferase enzymes and orthotopically implanted in the brains of NOD-SCID mice [36]. Therefore, our method is easily adaptable to other types of cancer and may be particularly interesting to dissect the molecular mechanisms driving the progression of rare subtypes of cancer where the analysis of human samples is complicated due to reduced incidence [57,58,59,60].

It is also noteworthy that a key characteristic of our method concerns its ability to distinguish luciferase-labelled cells, irrespective of admixture with unlabelled cells. As reported here, our method is equally efficient in detecting single luciferase-labelled cells, whether they are implanted singly or in combination with non-cancer cells. Similarly, this principle may also be applied to better characterize the interaction of unlabelled cancer cells co-implanted with luciferase-labelled non-cancer cells to further clarify the relevance of this crosstalk either in the TME or in pre-metastatic niches [35,61]. Furthermore, the versatility of detecting cell lines of both human (MDA-MB-231 and MCF7 cells) and murine (4T1 cells) origins expands the range of potential questions to be addressed by this method, as xenograft and allograft models may be chosen according to need. This approach may greatly increase our understanding of the role played by stromal cells, as demonstrated here for CAFs, but also expanded to investigate the contribution of vascular cells and adipocytes [50,51,62,63]. Additionally, the potential of using syngeneic models is remarkably interesting for the study of immune cells and therapies targeting immune components, as the implementation of humanized mouse cancer models is still challenging for many laboratories [64].

Notably, numerous luciferase enzymes and mutant variants of such enzymes have been described over the past few years [65,66,67,68,69]. Such variability expands the application of bioluminescence-based techniques because the activity of different luciferase enzymes may often be distinguished by substrate specificity or by the wavelength of the emitted light. Gaussia luciferase, for example, oxidizes coelenterazine and emits light in the blue spectrum [70], while Firefly luciferase oxidizes firefly luciferin (D-luciferin) and emits light closer to the red spectrum [71]. Provided that such luciferase enzymes are distinguishable, different luciferases may be used to label cancer cells and non-cancer cells and used in vitro and in vivo in models of direct coculture and co-implantation. Moreover, such bioluminescent enzymes may be used in a dual labelling system to gain additional insight relevant to metastatic progression. Our group has been a pioneer in this field and engineered cancer cells to constitutively express Gaussia luciferase and conditionally express Firefly luciferase in response to either BMP signaling activation [37] or TGF-β signaling activation [38]. Using this dual labelling model, D-luciferin injections offer a substrate that is specifically used for the quantification of changes in signaling pathway activity in cancer cells, further allowing continued monitoring of such changes over time in live mice [37,38,72,73,74]. Yet, as these cells are also labelled to constitutively express Gaussia luciferase, ex vivo analyses such as those reported in the current study allow the detection and quantification of disseminated cancer cells in tissue and organ samples injected with coelenterazine [37,38].

Another issue addressed by the model described here concerns the limited availability of samples when studying cancer progression. Although sample availability is not as limiting for pre-clinical cancer studies as it is for most clinical studies, it certainly imposes some restrictions. This is particularly problematic for the longitudinal assessment of cancer progression through the detection of shedded CTCs. Quantifying and analysing CTCs in blood samples from cancer patients has been repeatedly proposed as an efficient strategy to investigate and monitor cancer progression [6,75]. Nevertheless, CTCs are heterogeneous and rare [76]. Consequently, detecting and quantifying human CTCs remain a complex task and often relies on large volumes of blood sample [76,77]. Investigating CTCs in murine cancer models is further complicated as the repeated harvesting of large blood volumes may impact animal welfare, compromising the continuity of the study and the reliability of the collected data. In this scenario, it is noteworthy that our model enables the detection of CTCs in as little as five microliters of blood samples [36,37,38], making it an attractive method to be used for the monitoring of cancer progression over time. In addition to minimising the number of required animals, another direct implication of such an approach is improving the robustness of the obtained data. Different from other models where several animals must be grouped and each group must be humanely killed at a given time point, our model enables continuous tracking of the same mouse until the experimental endpoint is reached.

## 5. Conclusions

This work establishes novel mouse models to analyse spontaneous breast cancer progression through accurate quantification of CTCs, multiorgan metastasis, and tumour self-seeding. Combined with fast and simple sample processing, the elevated levels of bioluminescence emitted by luciferase enzymes allow ex vivo detection of single luciferase-labelled cancer cells early after the re-seeding and colonization of metastatic sites. Because metastasis is a multistep process, the precise contribution of pro-metastatic signalling pathways may change at each step. Thus, dissecting the differences between each metastatic step is critical for the development of efficient antimetastatic therapies and paramount to increasing the survival of cancer patients.

## Figures and Tables

**Figure 1 cells-13-01166-f001:**
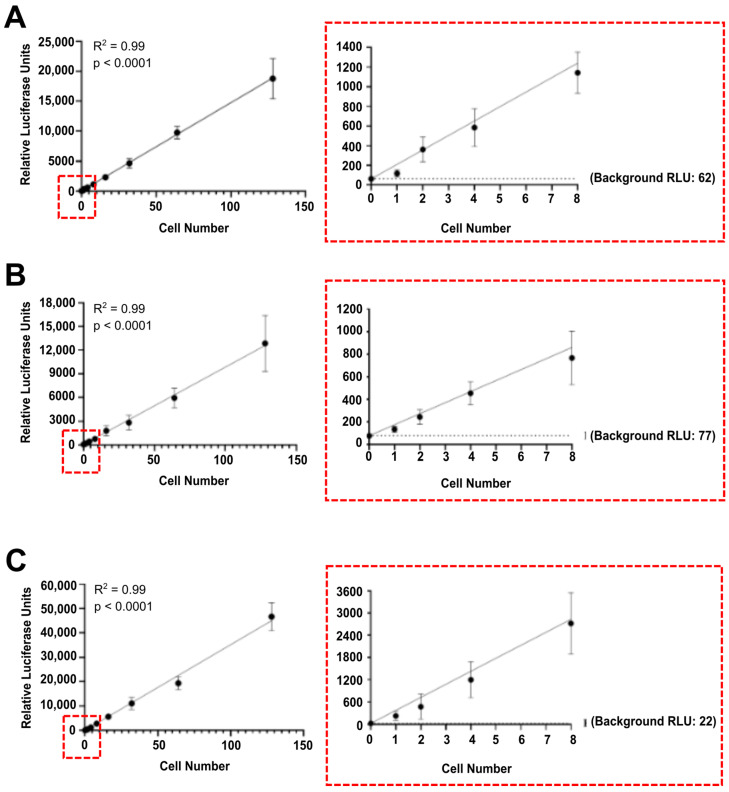
Luciferase activity is detected in vitro in single luciferase-labelled tumour cells. Quantification of luciferase-labelled tumour cells in PBS. (**A**) MDA.Gluc, (**B**) MCF7.Gluc, and (**C**) 4T1.Fluc cells resuspended in PBS were serially diluted to single cell and lysed before quantification of luciferase activity in cell lysates. Left panels show results obtained with all dilution points analysed. Right panels show enlarged regions of left panels, highlighting results obtained with 0–8 cells. Data are represented as mean ± SEM and corresponds to relative luciferase units (RLU) obtained from six readings. Linear correlation was determined by best-fit linear regression analysis.

**Figure 2 cells-13-01166-f002:**
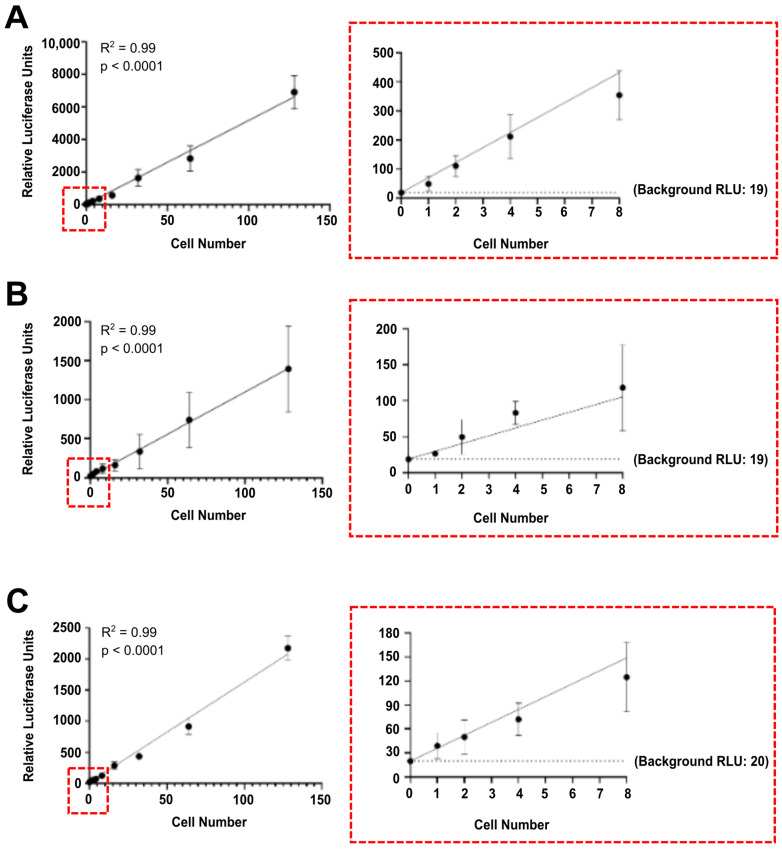
Tumour cells spiked in mouse blood and organ samples are detected at single cell level. Detection and quantification of (**A**) MDA.Gluc, (**B**) MCF7.Gluc, and (**C**) 4T1.Fluc cells spiked in mouse blood samples. Left panels show results obtained with all dilution points analysed. Right panels show enlarged regions of left panels, highlighting results obtained with 0–8 cells. Data are represented as mean ± SEM and corresponds to relative luciferase units (RLU) obtained from six readings. Linear correlation was determined by best-fit linear regression analysis.

**Figure 3 cells-13-01166-f003:**
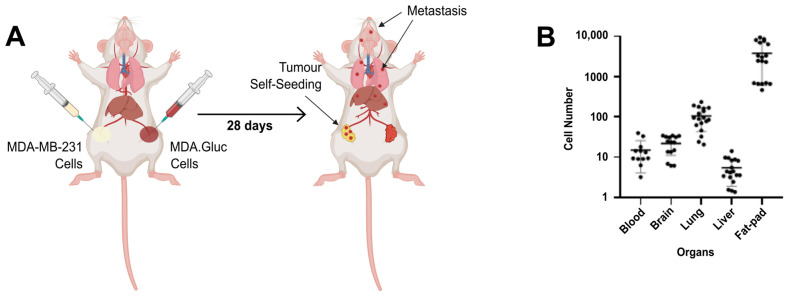
Luciferase-labelling enables the detection of CTCs, multiorgan metastasis and tumour self-seeding in an MDA-MB-231 orthotopic breast cancer model. (**A**) Schematic representation of the orthotopic and contralateral breast cancer mouse model used in this study. Created with BioRender.com accessed on 26 April 2024. 1 × 10^6^ wild type (unlabelled) MBA-MB-231 and 3 × 10^6^ MDA.Gluc cells were orthotopically inoculated in contralateral mammary fat pads. (**B**) Ex vivo quantification of luciferase activity in samples harvested 28 days post-implantation and after humane killing. Raw RLU values were normalized into cell numbers as described in the Materials and Methods section. Data are represented as mean ± SEM. Blood (n = 4), brain (n = 5), lung (n = 6), liver (n = 6) and mammary fat pad containing unlabelled tumour (n = 6) samples were analysed in triplicates.

**Figure 4 cells-13-01166-f004:**
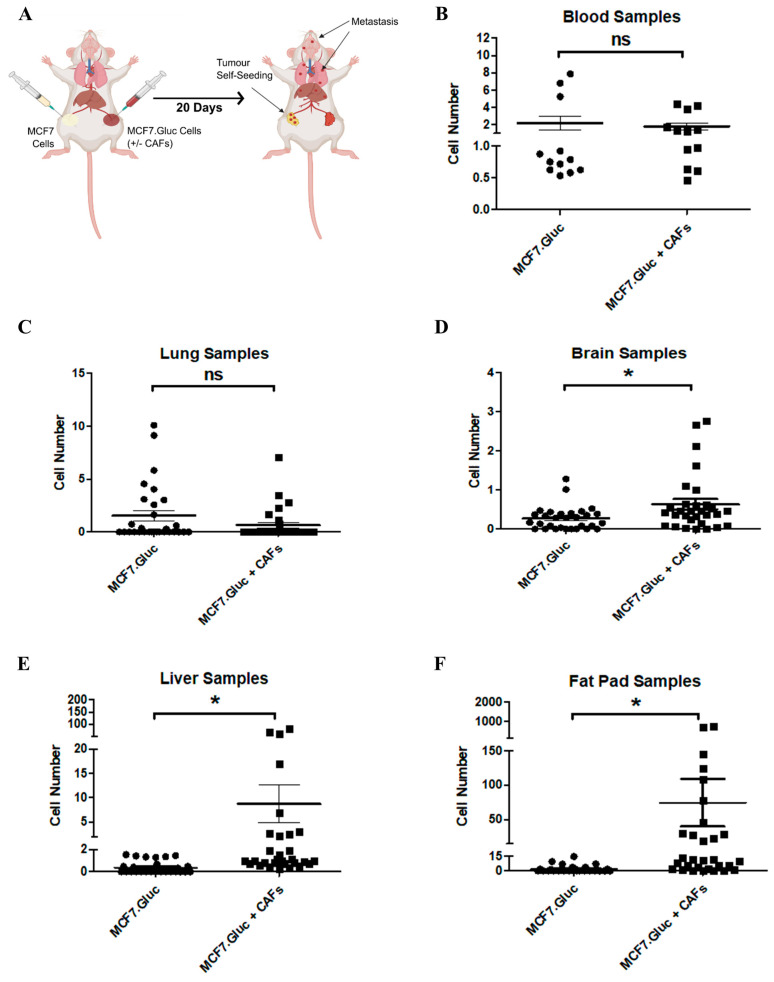
Luciferase-labelling allows ex vivo quantification of the dissemination of poorly metastatic MCF7 breast cancer cells. (**A**) Schematic representation of the orthotopic and contralateral breast cancer mouse model used in this study. Created with BioRender.com accessed on 26 April 2024. 5 × 10^6^ wild type (unlabelled) MCF7 and 5 × 10^6^ of MCF7.Gluc cells (±1.5 × 10^6^ CAFs) were orthotopically inoculated in contralateral mammary fat pads. (**B**–**F**) Ex vivo quantification of luciferase activity in (**B**) blood, (**C**) lung, (**D**) brain, (**E**) liver, and (**F**) mammary fat pads containing unlabelled tumour samples harvested 20 days post-implantation and after humane killing. Raw RLU values were normalized into cell numbers as described in the Materials and Methods section. Data are represented as mean ± SEM. Samples were harvested from six mice (n = 6). Blood samples were analysed in duplicates. Five brain, lung, liver, and mammary fat pad samples were analysed per mouse. ns: statistically non-significant, * *p* < 0.05.

## Data Availability

This study does not include data deposited in external repositories. All data supporting the findings of this study are available within the article and as Appendix A. Raw data and materials are available from the corresponding authors upon reasonable request.

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
