# Peer review of "A Novel Method for the Early Detection of Single Circulating, Metastatic and Self-Seeding Cancer Cells in Orthotopic Breast Cancer Mouse Models"

_cells, 2024, doi:10.3390/cells13141166_

Round 1
Reviewer 1 Report
Comments and Suggestions for Authors
In this manuscript, the authors described the methodology to develop a detection technique entitled "A novel method for the early detection of single circulating, metastatic, and self-seeding cancer cells in orthotopic breast cancer mouse models"
Comments
· The manuscript's author utilised a period of 20 days for co-implantation with CAFs, commencing as early as 20 days after the experiment's initiation. The authors are required to present data reflecting both long-term and early-stage co-implantation.
· The authors employed a concentration of 2mg/mL G418 sulphate for transfecting cells. They need to explain how this specific dosage was determined. The authors need to provide data on the effects of varying concentrations of G418 on the targeted cell line.
· The authors need to incorporate additional experiments to validate the detection methods, such as Immunofluorescence and Immunohistochemical Staining.
· Figures 4B, 4C, and 4D require clearer images and better labelling.
· In the results section, the authors state that colonies with the highest luciferase activity were selected, expanded, and utilised in subsequent analyses. However, the manuscript does not include any images of the colony-forming cells. The authors need to provide images of the colonies with and without luciferase expression.
· Twenty-eight days after tumour inoculation, the animals were humanely euthanised, and samples of blood and organs were collected. The luciferase activity in the sample lysates was then quantified. The authors should include images of the organs from both the control and treatment groups.
· The authors are advised to elaborate on the methodology, expand the discussion, and revise the conclusion.
· The authors need to include a section detailing the limitations of the study.
Comments on the Quality of English Language
Please see above section.
Author Response
Comment 1: The manuscript's author utilised a period of 20 days for co-implantation with CAFs, commencing as early as 20 days after the experiment's initiation. The authors are required to present data reflecting both long-term and early-stage co-implantation.
Response 1: Authors agree that the number of MCF7.Gluc cells detected in blood, brain, lung, mammary fat pad, and liver samples may differ if quantified at earlier and longer time points. However, our experiment was not designed to comprehensively analyse the metastatic dissemination of these breast cancer cells over time. Rather, this experiment was performed to investigate if the method presented in this manuscript could be used to detect the dissemination of MCF7.Gluc cells (Figure 4) as similarly done for MDA.Gluc cells (Figure 3). To better reflect our goal, the following was added: “To further assess the value of our method and validate its use in a different breast cancer model…” (p. 6 – lines 259-260).
Comment 2: The authors employed a concentration of 2mg/mL G418 sulphate for transfecting cells. They need to explain how this specific dosage was determined. The authors need to provide data on the effects of varying concentrations of G418 on the targeted cell line.
Response 2: The following was added to the section Establishing stable luciferase-labeled cell lines to clarify the procedure: “This concentration of G418 sulfate was established by treating breast cancer cell lines with increasing concentrations of G418 sulfate and visually determining the minimum concentration of this antibiotic that could kill all the cells after 72h in an antibiotic kill curve” (p.3 – lines 111-114).
Comment 3: The authors need to incorporate additional experiments to validate the detection methods, such as Immunofluorescence and Immunohistochemical Staining.
Response 3: Authors agree that immunostaining could be used to confirm the detection of metastatic clusters and nodules as traditionally been done. Nevertheless, it may not be suitable when investigating the initial stages that follow the seeding of disseminated tumour cells into secondary sites. At this stage, few metastatic cells are expected to exist and their distribution throughout the targeted organ is scarce. Therefore, as we aimed at detecting early single disseminated cancer cells, immunostaining cannot be used as a validating method.
Comment 4: Figures 4B, 4C, and 4D require clearer images and better labelling.
Response 4: Apparently, there was a problem with typesetting, which was already corrected by the journal’s team. According to the assistant editor: “…we noticed that the reviewer mentioned Figure 4. We are really sorry for the mistake we made in typesetting the manuscript. We have updated the correct version to the system”.
Comment 5: In the results section, the authors state that colonies with the highest luciferase activity were selected, expanded, and utilised in subsequent analyses. However, the manuscript does not include any images of the colony-forming cells. The authors need to provide images of the colonies with and without luciferase expression.
Response 5: Authors have rephrased this description to better reflect the procedure as the term ‘colony’ may be misleading when the intention was to communicate clonal selection by standard protocol (PMID: 21913800, PMID: 38105247, PMID: 38275817). Please see the following: “Specifically, single cells were seeded in 96-well plates (0.5 cell/well), clones were grown independently and evaluated for luciferase activity. Clones with the highest luciferase activity were selected, expanded, and used in following analyses” (p. 4, lines 181-184).
Comment 6: Twenty-eight days after tumour inoculation, the animals were humanely euthanised, and samples of blood and organs were collected. The luciferase activity in the sample lysates was then quantified. The authors should include images of the organs from both the control and treatment groups.
Response 6: Short-term in vivo experiments presented in this manuscript were designed to detect early stages of metastasis when macroscopic metastatic nodules are not yet visible. Considering the lack of morphological changes, organs were fragmented and further processed as described in the manuscript without previous imaging.
Comment 7: The authors are advised to elaborate on the methodology, expand the discussion, and revise the conclusion.
Response 7: Changes were made as proposed to improve the understanding of the study. Please see highlighted text throughout the revised manuscript.
Comment 8: The authors need to include a section detailing the limitations of the study.
Response 8: Authors agree with importance of including study limitations and they were presently included in the discussion section as suggested by the reviewer. Please see highlighted text in the revised manuscript (e.g., page 11, lines 340-343, page 11, lines 344-350, page 12- lines 386-390).
Reviewer 2 Report
Comments and Suggestions for Authors
In this work, the authors describe a novel method to study the metastatic process, analyzing the bioluminescence at a single cell level using luciferase transfected cells and detected in blood and different organs.
Comments:
1- The authors obtained the transfected 4T1 cells and did the in vitro experiments, but they do not show the in vivo experiments with these cells. It will be interesting that if they performed these experiments, to mention the results. If they did not, to mention if it is planed, as a confirmation of the findings in a different model.
2- FIGURE 1: the legends B and C in the figure are missing.
3- FIGURE 4: the pictures A, B, C and D are overlapped one to each other.
4- It will be interesting if the authors show the metastasis in the different organs by histology, to confirm that the cells observed by luciferase activity give a stablished metastasis, particularly in the MCF7 model which is known that is a non-metastatic cell line.
Author Response
Comment 1: The authors obtained the transfected 4T1 cells and did the in vitro experiments, but they do not show the in vivo experiments with these cells. It will be interesting that if they performed these experiments, to mention the results. If they did not, to mention if it is planed, as a confirmation of the findings in a different model.
Response 1: The 4T1 model has not been comprehensively used for in vivo analyses yet. To emphasize the point suggested by the reviewer, we have changed the discussion as follows: “Noteworthy, the metastatic dissemination of 4T1 cells was not investigated in vivo in the current study. However, this effort in expanding the diversity of existing models of cancer metastasis may enable future studies that aim at investigating breast cancers that are distinguished, for example, by metastatic potential and response to immune therapies” (p. 12, lines 386-390).
Comment 2: FIGURE 1: the legends B and C in the figure are missing.
Response 2: This problem has been addressed as highlighted in the Figure 1 for panels B and C in the revised manuscript.
Comment 3: FIGURE 4: the pictures A, B, C and D are overlapped one to each other.
Response 3: Apparently, there was a problem with typesetting that was already corrected by the journal. According to the assistant editor: “…we noticed that the reviewer mentioned Figure 4. We are really sorry for the mistake we made in typesetting the manuscript. We have updated the correct version to the system”.
Comment 4: It will be interesting if the authors show the metastasis in the different organs by histology, to confirm that the cells observed by luciferase activity give a stablished metastasis, particularly in the MCF7 model which is known that is a non-metastatic cell line.
Response 4: Authors agree that histological analysis would be useful to validate large clusters of metastatic cells and macrometastasis. However, such strategy may not be proportionally accurate in our model as the current study focus on reporting a method capable of detecting early stages of the metastatic cascade, i.e., soon after the re-seeding of disseminated cancer cells when metastatic cells colonizing secondary sites are expectedly scarce.
Reviewer 3 Report
Comments and Suggestions for Authors
The manuscript submitted by Murad et al. titled “A novel method for the early detection of single circulating, metastatic and self-seeding cancer cells in orthotopic breast cancer mouse models” authors study the metastatic potential of breast cancer cells in two models, MDA-MB-231 and MCF7 cells. Authors showed bioluminescence at single cell level in blood, brain, lung, liver, and mammary fat pad samples. Ex vivo, circulating tumor cells, metastasis, and tumor self-seeding were detected from animals implanted with highly metastatic luciferase-labelled MDA-MB-231 cells. Authors also tested and showed that metastatic luciferase labelled MCF7 cells were limited and increased upon co-implantation with CAFs.
The use of bioluminescence at single cell level is novel and significant as the authors have already shown and published the method in brain cancer model (Cells 2024 Jan 19;13(2):192. doi: 10.3390/cells13020192). How these studies will further add to the advantage of bioluminescence in detection of single cells is not clear in the manuscript.
I have some major critiques:
Why authors have not implanted the cells with Matrigel in orthotopic breast cancer model which would increase the physiological relevance of the model. MCF-7 cells tumors can metastasize to lungs, liver, and spleen. Since the cell line is hormone dependent, the delay in metastasis may be due to the lack of hormone and 17β-estradiol treatment increases both the growth rate and frequency of metastases.
The rationale for selecting the number of cells for tumor injections is not clear in the manuscript. Why 3 million luciferase-labelled MDA-MB-231 cells were injected at one site while only one million unlabeled MDA-MB-231 cells were injected.
The rationale for implanting 5 million labelled MCF7 cells (± 1.5 million CAFs) and 5 million wild type (unlabeled) MCF7 cells is not clear in the manuscript.
The endpoints, why 28 days in MDA-MB-231 cells and 20 days in MCF7 cells is not clear in the manuscript. Time course experiment would have added to the physiological relevance of the model.
There are no studies performed to check the markers of metastasis at different sites of metastasis. To metastasize, cancer cells break off from the primary tumor and travel through the blood or lymph to other organs. Why lymph nodes were not included in the manuscript is not clear.
Overall, the article is well written, but additional details are needed in results and methods.
Comments on the Quality of English LanguageMinor English Revision
Author Response
Comment 1: Why authors have not implanted the cells with Matrigel in orthotopic breast cancer model which would increase the physiological relevance of the model. MCF-7 cells tumors can metastasize to lungs, liver, and spleen. Since the cell line is hormone dependent, the delay in metastasis may be due to the lack of hormone and 17β-estradiol treatment increases both the growth rate and frequency of metastases.
Response 1: Authors appreciate the metastatic potential of MCF7 cells highlighted by the reviewer and agree that implanting these cells with Matrigel would likely increase their dissemination. In the context of the method reported in this study, however, the metastatic spreading of MCF7 cells were analyzed primarily to validate that single circulating, metastatic and self-seeding cells are detectable by ex vivo luciferase assay and this is not exclusively applied to MDA-MB-231 cells. To emphasize the relevance of the suggestion made by the reviewer that could greatly contribute to mechanistic studies, the following was added: “Noteworthy, such results were achieved only 20 days after the simple inoculation of cancer cells resuspended in saline solution, exhibiting great potential to be further optimized by using a Matrigel solution that would offer additional support to cancer cells at the time of the implantation” (p. 11, lines 340-343).
Comment 2: The rationale for selecting the number of cells for tumor injections is not clear in the manuscript. Why 3 million luciferase-labelled MDA-MB-231 cells were injected at one site while only one million unlabeled MDA-MB-231 cells were injected.
Response 2: The following was added to clarify the difference highlighted by the reviewer: “Cell numbers and experiment duration were defined considering our previous studies [38,39] as to compensate for the differences in the growth of wild type and luciferase-labelled cells” (p. 5, lines 230-232).
Comment 3: The rationale for implanting 5 million labelled MCF7 cells (± 1.5 million CAFs) and 5 million wild type (unlabeled) MCF7 cells is not clear in the manuscript.
Response 3: This issue was addressed as done for the previous point by adding the following: “The duration of this experiment and the number of cells to be orthotopically inoculated considered our previous experience [38,39] and were defined to compensate the growth of labelled and unlabeled tumours” (p. 6, lines 265-268).
Comment 4: The endpoints, why 28 days in MDA-MB-231 cells and 20 days in MCF7 cells is not clear in the manuscript. Time course experiment would have added to the physiological relevance of the model.
Response 4: The timelines chosen for in vivo experiments including MDA-MB-231 and MCF7 cells were also defined based on previous studies done by our group that validated metastatic dissemination at these time points (please see the modified text in p. 5, lines 230-231 and p. 6, lines 265-268). Authors appreciate that a time course would add valuable information about MDA-MB-231 and MCF7 metastasis. Yet, we would like to reiterate that the main aim of this manuscript was not to comprehensively characterize the metastatic spreading of these cells, but rather to demonstrate that single disseminated breast cancer cells are detectable by ex vivo luciferase assay. Thus, authors understand that results obtained with a time course would not change the feasibility of using the proposed method.
Comment 5: There are no studies performed to check the markers of metastasis at different sites of metastasis. To metastasize, cancer cells break off from the primary tumor and travel through the blood or lymph to other organs. Why lymph nodes were not included in the manuscript is not clear. Overall, the article is well written, but additional details are needed in results and methods.
Response 5: Authors agree that markers could be investigated but while such analyses would be welcome to detect larger impacts caused metastasis outgrowth, they would show lower accuracy when only few metastatic cells have re-seeded in the metastatic niche – the specific context of our study. We also appreciate the relevance of lymph vessels to this outcome and whereas hematogenous dissemination was the focus of this current manuscript, additional investigation must be done in the future to understand the feasibility of using ex vivo luciferase assay to analyse lymph nodes. The importance of additional studies investigating the colonization of lymph nodes was acknowledged in the revised manuscript as follows: “The successful establishment of metastases involves a sequence of events that includes the invasion of cancer cells at the primary site, their dissemination, the colonization of secondary sites, and the outgrowth of metastatic lesions. Although this study focused on investigating blood-borne metastases, it is important to note that the intravasation and dissemination of cancer cells via lymph vessels may also account for a significant part of the metastatic lesions and other organs (e.g., lymph nodes) shall be investigated in future studies” (p. 11, lines 344-350).
Round 2
Reviewer 3 Report
Comments and Suggestions for Authors
The authors revised the manuscript and I recommend it to acceptable to publish in Cells.